# Explicit and implicit timing in older adults: Dissociable associations with age and cognitive decline

**Mariagrazia Capizzi**[1ʘ]\*, **Antonino Visalli**[2ʘ], **Alessio Faralli**[3], **Giovanna Mioni**[4]\*

**1** Univ Paul Valéry Montpellier 3, Montpellier, France, **2** Department of Neuroscience, University of Padova, Padova, Italy, **3** Department of Neuroscience, Psychology, Drug Research, and Child Health (NEUROFARBA), University of Florence, Florence, Italy, **4** Department of General Psychology, University of Padova, Padova, Italy

ʘ These authors contributed equally to this work.
\* giovanna.mioni@unipd.it (GM); mgcapizzi@hotmail.com (MC)

**Data Availability Statement:** All relevant data are available from Open Science Framework database (https://osf.io/sh492/).

**Funding:** The author(s) received no specific funding for this work.

## Abstract

This study aimed to test two common explanations for the general finding of age-related changes in the performance of timing tasks within the millisecond-to-second range intervals. The first explanation is that older adults have a real difficulty in temporal processing as compared to younger adults. The second explanation is that older adults perform poorly on timing tasks because of their reduced cognitive control functions. These explanations have been mostly contrasted in *explicit* timing tasks that overtly require participants to process interval durations. Fewer studies have instead focused on *implicit* timing tasks, where no explicit instructions to process time are provided. Moreover, the investigation of both explicit and implicit timing in older adults has been restricted so far to healthy older participants. Here, a large sample (N = 85) comprising not only healthy but also pathological older adults completed explicit (time bisection) and implicit (foreperiod) timing tasks within a single session. Participants' age and cognitive decline, measured with the Mini-Mental State Examination (MMSE), were used as continuous variables to explain performance on explicit and implicit timing tasks. Results for the explicit timing task showed a flatter psychometric curve with increasing age or decreasing MMSE scores, pointing to a deficit at the level of cognitive control functions rather than of temporal processing. By contrast, for the implicit timing task, a decrease in the MMSE scores was associated with a reduced foreperiod effect, an index of implicit time processing. Overall, these findings extend previous studies on explicit and implicit timing in healthy aged samples by dissociating between age and cognitive decline (in the normal-to-pathological continuum) in older adults.

## Introduction

Age-related changes in the performance of timing tasks within the millisecond-to-second range intervals are commonly reported [1–6]. Performance of older adults on timing tasks has been mostly tested using *explicit* timing tasks, which overtly inform participants about the

**Competing interests:** The authors have declared that no competing interests exist.

temporal nature of the task [7]. For instance, in the time bisection task (the one also used in the present study as illustrated in Fig 1A), participants first learn a "short standard" and a "long standard" duration and then classify intermediate durations as being more similar to the short or the long standard [8, 9].

According to influential pacemaker-based models of time perception [10], performance on the time bisection task relies on the functioning of both internal "clock" and memory/decision stages. The internal clock is conceived of as a pacemaker-counter device that emits pulses accumulated in a counter (i.e., the greater the number of pulses, the longer the estimation of the interval duration). The pulses stored into the accumulator are then transferred to working memory; an additional decision stage finally compares the pulses accumulated in working memory to those already stored in a reference memory system (i.e., the short and long standards for the time bisection task) to identify an appropriate outcome.

Building up on pacemaker-based models of time, it is thus debated whether the age-related changes observed in explicit timing tasks can be genuinely attributed to a dysfunction at the level of the clock stage, hereafter referred to as "temporal processing" (e.g., a slower clock accumulating less pulses in older than younger adults), or should be rather considered as a secondary deficit at the level of memory/decision stages, hereafter referred to as "cognitive control functions" (e.g., a noisier memory representation of the short and long standards in older adults; see Fig 1B for a depiction of the behavioural patterns usually obtained in the time bisection task for both ideal and possible age-related performances). Support for a secondary cognitive deficit of older adults in timing tasks also comes from studies of pathological aged populations such as, for example, patients with Alzheimer's disease (AD). Relative to age-matched controls, AD patients–who are known to present severe cognitive deficits [11, 12]–perform poorly on explicit timing tasks (see [13, 14] for reviews). As will become clear below, the main goal of the present study on older adults was to elucidate the role of age and cognitive decline in explicit and implicit timing tasks.

Explicit requirements to memorize or to pay attention to interval durations are instead absent in the context of *implicit* timing tasks, hence, making these tasks less demanding with respect to explicit timing ones. As illustration, consider a warned reaction time (RT) task (like the task used here as depicted in Fig 2A) that just entails a fast response to a target stimulus. The time interval between the warning signal (thicker circle) and the target (cross symbol), otherwise known as "foreperiod", could assume one of different durations with equal a-priori probability at the beginning of each trial. In this kind of tasks, the probability that the target will occur at the longest foreperiods grows up with the passage of time, as formally described by the hazard function, i.e., the conditional probability that an event will occur given it has not yet occurred ([15–17]; see Fig 2B, for an illustration of the hazard rate). Participants are sensitive to the hazard function, a conclusion supported by their shorter RTs for the longer foreperiod trials, the so-called "variable foreperiod effect", hereafter simply referred to as the foreperiod effect [18–21], which is represented by the back line in Fig 2B. The behavioural benefit afforded by elapsing time during longer foreperiod trials is interpreted as evidence for an implicit processing of time, since the foreperiod effect occurs even if participants are not explicitly instructed to pay attention to time or are uninformed about the used interval durations.

To our knowledge, explicit and implicit timing have been thus far compared in healthy older adults only [2, 22]. As an example, Droit-Volet and colleagues [2] devised a between-participants design in which one group of older adults and one group of younger adults performed an explicit timing task (i.e., temporal generalization task), whereas different groups of older and younger adults were engaged in an implicit timing task (i.e., a variant of the foreperiod task; [23]). Participants' performances on explicit and implicit timing tasks were also

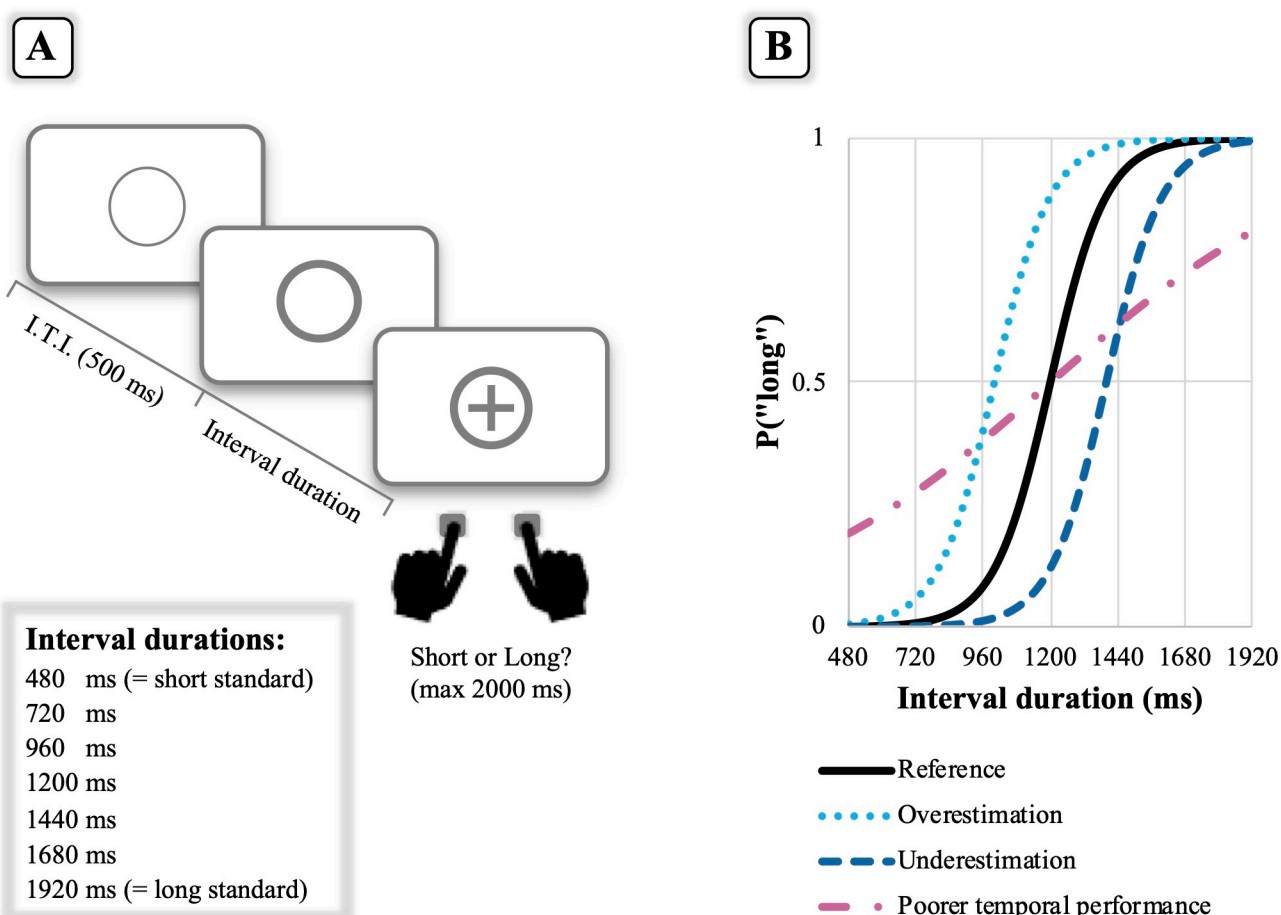

**Fig 1. (A) Schematic view of the explicit (time bisection) task.** In a first training phase, participants memorized a "short standard" (480 ms) and a "long standard" duration (1920 ms). In a subsequent testing phase, they indicated whether the interval duration elapsing from the onset of the thicker circle to the onset of the cross was closer to the previously memorized "short standard" or "long standard". Responses were given by pressing two response keys on the computer keyboard. I.T.I. stands for Inter-Trial-Interval. **(B) Patterns of behavioural performance in the time bisection task.** A common way of looking at performance on the time bisection task is to construct a psychometric curve by plotting the proportion of trials in which participants respond "long" as function of interval duration. The black psychometric curve depicts an ideal performance in which participants never respond "long" to the short standard duration, whereas they always respond "long" to the long standard. At the intermediate duration (1200 ms), they are equally likely to respond short or long. Relative to the reference, a shift of the psychometric curve towards the left (light blue line) or the right (blue line) means over- or under-estimation, respectively. Conversely, a flatter psychometric curve (pink line) is indicative of a poorer temporal performance (i.e., participants tend to respond "short" to long durations and "long" to short durations). Although it is difficult to completely isolate clock (i.e., "temporal processing") from memory/decision stages (i.e., "cognitive control functions"), it makes sense to hypothesize that age-related changes in clock speed should be mainly expressed by a rightward shift of the psychometric curve (i.e., a slower clock in older adults). By contrast, a flatter psychometric curve in older adults could be likely attributed to a deficit in the additional cognitive control functions (e.g., working memory) thought to be required to correctly perform on the time bisection task.

correlated to cognitive scores derived from neuropsychological tests. Results showed that older adults were as accurate as younger adults in both explicit and implicit timing tasks; however, older participants were more variable than younger ones in the explicit timing task and their performance was explained by lower attentional capacity rather than age. By contrast, older adults showed a greater reliance on the hazard function than younger adults, a result that was significantly associated with age but not cognitive scores.

Collectively, the correlational findings by Droit-Volet and colleagues [2] speak to different possible influences of age and cognitive functions in explicit and implicit timing tasks. In the present study, we aimed to advance our knowledge about performance of older adults on

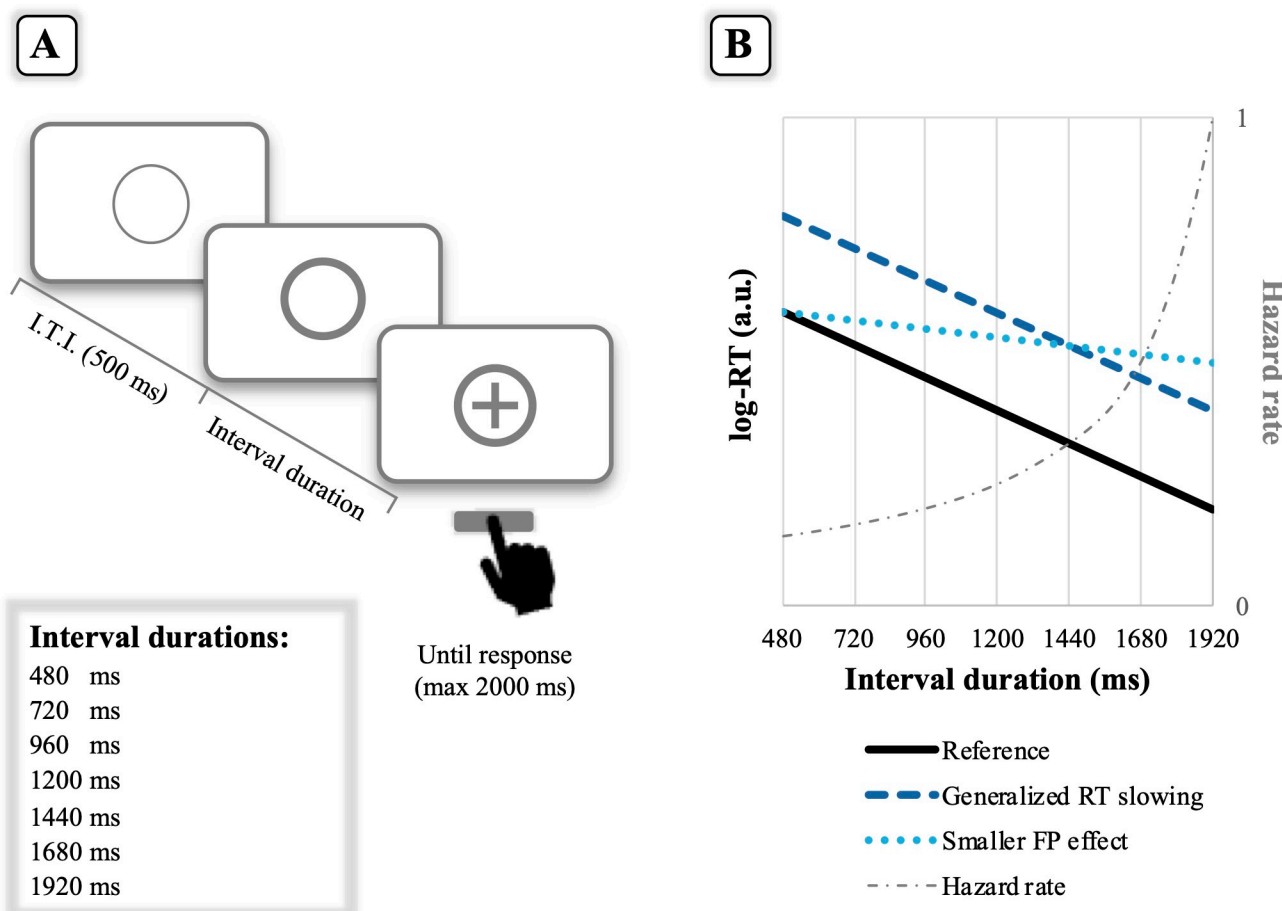

**Fig 2. (A) Schematic view of the implicit (foreperiod) task.** The foreperiod task comprised the same stimulus material and general procedure of the time bisection task, but differed in the specific task instructions given to participants. Specifically, the participants' task was to press the spacebar as quickly as possible whenever the cross appeared inside the circle. Thus, no instructions to memorize interval durations were provided for the implicit timing task. The interval duration (or foreperiod, FP) separating the thicker circle from the cross could randomly assume one of seven values with equal a-priori probability. I.T.I. stands for Inter-Trial-Interval. **(B) Patterns of behavioural performance in the foreperiod task.** Performance on the foreperiod task is plotted in terms of reaction time (RT) as function of interval duration. For illustrative purposes, the depicted log-RTs are in arbitrary units (a.u.). The black line shows the reference finding typically observed in the foreperiod task, with shorter RTs at longer interval durations (i.e., the foreperiod effect). The foreperiod effect is formally described by the hazard function (represented in grey), that is, the increasing conditional probability that an event will occur given it has not yet occurred. The size of the foreperiod effect is taken as evidence for an implicit processing of time, considering that participants are not explicitly instructed to memorize or use interval durations. Relative to the reference, the blue line depicts the presence of a generalized RT slowing that, however, does not imply a deficit in the processing of implicit timing, as indexed by the size of the foreperiod effect. By contrast, the light blue line depicts a smaller foreperiod effect, which is indicative of an impaired use of implicit timing (i.e., no benefit afforded by the passage of time). Assuming that implicit timing tasks rely less on cognitive resources than explicit timing tasks leads to the prediction that processing of implicit timing should be spared in older adults as compared to processing of (more demanding) explicit timing (i.e., a pattern more consistent with the blue, rather than the light blue, line).

explicit and implicit timing tasks by considering not only healthy but also pathological older participants. This allowed capturing differences in explicit and implicit timing tasks linked to age and pathological cognitive decline, two variables that, although often correlated, are not systematically associated [24]. To this end, healthy older adults and individuals diagnosed with either Mild Cognitive Impairment (MCI) or dementia completed explicit (time bisection) and implicit (foreperiod) timing tasks in a single session. The Mini-Mental State Examination (MMSE; [25]) was used as an index of cognitive decline. The MMSE represents, indeed, one of the most routinely used screening tools in clinical practice, even if it provides only a generic

assessment of cognitive decline. Of note, although unhealthy participants received a formal diagnosis of (mild-) cognitive dementia (see the Methods Section), hereafter any reference to cognitive impairment/decline in our sample is related to the MMSE score. Participants' ages and MMSE scores were considered as continuous predictors of performance in the analyses.

Concerning implicit timing, we expected to find a significant association of performance on the foreperiod task with age rather than MMSE scores, replicating previous research [2]. As concerns explicit timing, if the poor performance of older participants on the time bisection task depends on a deficit in temporal processing, this should be reflected by a (rightward) shift of the psychometric curve. According to previous literature reporting a slowing down of the pacemaker with age (see [26]), we predicted to find a significant association between the rightward shift of the curve and age. If, conversely, changes in the performance of older participants depend on their reduced cognitive control functions, this should translate into a flatter psychometric curve. Therefore, we predicted a significant association between MMSE scores and the flattening of the curve.

## Method

### Participants

Ninety-one older adults recruited from different centers in the Italian territory voluntarily took part in the study. Six participants were excluded as they just completed the implicit timing task leaving a final sample of 85 older adults (this sample allows observing a correlation with a Pearson's r of .3 with a power of .8). Twenty-three participants resided in the municipality of Padova; 14 were tested at their home and 9 at local social centers. Thirteen participants resided in the municipality of Sacile (Pordenone) and were tested at local social centers. Fifteen participants resided in the municipality of Vicenza and were tested at local social centers. Twenty-one participants resided in the municipality of Grosseto and were tested either at home (N = 14) or at a local social center (N = 7). Finally, 13 resided in the municipality of Cagliari and all were tested at home. Unhealthy participants received a diagnosis of either MCI or dementia by expert clinicians. To control for differences in the testing context (home vs. public centers), particular attention was given to the experimental setting such that all of the participants performed the temporal tasks in a quiet and normally illuminated room and all of them received the same instructions. Participants' cognitive decline was defined according to the score (corrected for age and education) obtained on the Italian version of the Mini-Mental State Examination (MMSE, [25]; [27], for the Italian version). Of note, although in the below analyses MMSE scores were considered as a continuous variable, for completeness S1 Table also reports the clinical classifications (i.e., dementia or MCI) of our sample as commonly done according to the cut-offs used in the literature, in addition to the demographic characteristics of the participants included in this study.

All the participants had normal or corrected-to-normal vision and normal hearing. All of them signed an informed consent before participation, in accordance with the Declaration of Helsinki. The experiment was approved by the Ethics Committee of the Department of General Psychology of the University of Padova (protocol n. 3387).

### Procedure and task

Participants were seated in a quiet room at an approximate distance of 60 cm from the computer screen (15.6") that produced and recorded experimental events via Psycophy Software [28]. Explicit and implicit timing tasks comprised the same stimulus material and general procedure but differed in the specific task instructions given to participants (Figs 1A and 2A). For both tasks, stimuli consisted of a grey circle and a grey cross presented at the center of a lighter

grey background screen. A thin circle was initially displayed for 500 ms (Inter-Trial-Interval, ITI), followed by a thicker circle that could assume one of the following interval durations: 480, 720, 960, 1200, 1440, 1680, or 1920 ms. After the duration had elapsed, a cross appeared in the center of the circle for 500 ms. For the *explicit timing task*, the experimental session started with a training phase, in which participants were instructed to memorize two standard durations: 480 (short standard) and 1920 ms (long standard), each presented 10 times. During a subsequent testing phase, participants had to indicate whether the temporal interval elapsing from the onset of the thicker circle to the onset of the cross was closer in duration to the previously memorized "short standard" or "long standard". Responses were given by pressing two response keys ("S" and "L" on the computer keyboard), which were covered with the labels "B" and "L" (i.e., "Breve" and "Lungo", respectively, meaning short and long in Italian); response keys were counterbalanced between participants.

In the *implicit timing task*, participants were instructed to press the spacebar as fast as possible whenever the cross appeared inside the thicker circle.

For both explicit and implicit timing tasks, no information about stimulus durations was given to participants. The experiment consisted of a total of 6 blocks (3 blocks for each timing task) of 42 trials each (6 repetitions for each temporal interval). Forty-four participants started with the explicit timing task, whereas 41 participants started with the implicit timing task. Explicit and implicit timing tasks were separated by a short break to allow participants a brief rest before undergoing the second task.

## Statistical analyses

The same statistical approach was applied to the analysis of both explicit and implicit timing tasks by means of Mixed-Effects models, which were implemented in the R environment (http://www.R-project.org/) using functions from the *lme4* library [29]. Concerning the explicit timing task, the probability of "long" responses was modelled through logistic regressions conducted with the *glmer* function (i.e., a generalized linear mixed model, GLMM, with probit-link function). Data from trials with missing responses were discarded from the analysis. The GLMM included "Interval duration", "MMSE score", "Age" variables and their interactions as fixed terms. The correlation between Age and MMSE score variables was very low (r = -.068). These continuous variables were centered and scaled to improve their interpretation and the fit of the model. Participants were treated as random effects. For the interpretation of the model terms, a significant main effect of MMSE score would indicate a change in the intercept value (since all variables were centered, the intercept is the expected value of the logistic curve at the middle interval duration, i.e., 1200 ms, when MMSE and AGE variables are at their mean value) for a 1 unit change in the MMSE score. As can be appreciated from Fig 1B, the higher the value of the psychometric curve at the middle interval duration, the higher the shift of the curve towards the left (i.e., over-estimation), and vice versa. In sum, a significant main effect of MMSE with an odds ratio greater than 1 would indicate a progressive shift of the curve towards the left with increasing MMSE score (if lower than 1, this would indicate a progressive shift of the curve towards the right with increasing MMSE score). The same logic applies to the main effect of AGE. The flattening of the curve represented in Fig 1B is captured by the interaction of MMSE score (or Age) and Interval duration. A significant odds ratio greater than 1 would indicate a significant steeping of the curve with the increase of the MMSE variable (or Age), whereas a significant odds ratio lower than 1 would indicate a significant flattening of the curve with the increase of the MMSE variable (or Age).

For the implicit timing task, linear regressions were conducted on log-transformed reaction times (RTs) by using the *lmer* function (i.e., a linear mixed model, LMM). As for the GLMM, a

full LMM was specified including "Interval duration", "MMSE score", "Age" variables and their interactions as fixed terms. These continuous variables were centered and scaled to improve their interpretation and the fit of the model. Participants were treated as random effects. Data from error trials, i.e., anticipated (< 100 ms) or missing responses to the target, were not included. For the interpretation of the model terms, a significant interaction between MMSE (or Age) and Interval duration would indicate changes in the foreperiod effect associated with the MMSE score (or Age). As explained above, the foreperiod effect is the well-observed lowering of RT with increasing interval duration. This effect is captured by the negative slope of the regression line (see Fig 2B). A significant negative interaction effect would indicate, hence, a stronger foreperiod effect with increasing MMSE score (or Age). On the contrary, a significant positive interaction would indicate a progressive reduction of the foreperiod effect with increasing in the variable (MMSE or Age).

To quantify and evaluate the contribution of MMSE scores and Age in explaining the data (and, hence, to justify their inclusion in the model), two model comparisons (as implemented in the lme4 function *anova*) were conducted, each one including three models: (i) a simple model with just Interval duration as fixed term; (ii) a model adding in one case MMSE scores and in the other case Age (and their interaction with Interval duration); and (iii) the full model including Interval duration and both MMSE score and Age variables.

Analyses were conducted on data from all the participants (N = 85) with no exclusion criteria. However, to check for the robustness of our results, we also repeated the above-mentioned analyses by excluding participants according to the proportion of trials in which they did not provide a response. Specifically, for each participant and separately for each task, we calculated the proportion of non-given responses. Then, the highest proportion of non-given responses between tasks was taken for each participant, and analyses were repeated four times including participants with a proportion of non-given responses lower than .1, .2, .3, or .4, respectively. Overall, these control analyses confirmed the robustness of our main findings (see S2 and S3 Tables).

## Results

### Explicit timing task

Model comparison (Table 1) showed that the best fitting model was the full model including Interval duration, MMSE and Age variables (a summary of the model output is presented in Table 2). Fig 3 shows the finding of significant interactions between Interval duration and MMSE and between Interval duration and Age. Specifically, the shape of the psychometric curve became flatter with decreasing MMSE scores and increasing age. The main effects of MMSE and Age, which indicate differences at the middle interval duration, were not significant, implying that there were no systematic changes in temporal judgements (i.e., over- or under-estimation) modulated by MMSE or Age.

### Implicit timing task

As for the explicit timing task, model comparison (Table 1) showed that the best fitting model was again the full model including Interval duration, MMSE and Age variables. Visual inspection of the residuals showed that they were skewed. Following Baayen and Milin [30], trials with absolute standardized residuals higher than 2.5 SD were considered outliers and removed (3% of the trials). After removal of outlier trials, the full model was refitted achieving reasonable closeness to normality. A summary of the model output is presented in Table 3. Fig 4 shows an overall increase in RT with decreasing MMSE scores (MMSE main effect) and with increasing Age values (Age main effect). Concerning the foreperiod effect (i.e., shorter RTs

**Table 1. Model comparison for the explicit and implicit timing tasks showed that the best fitting model was the full model including the interval duration and both the MMSE score and age variables.**

| Fixed effects | log-likelihood | χ2 (df) | p(>χ2) | AIC |
|---|---|---|---|---|
| *Explicit timing task (a)* | | | | |
| Interval duration | -8262.7 | | | 16531 |
| Interval duration × MMSE | -8166.9 | 191.7(2) | < .001 | 16344 |
| Interval duration × MMSE × Age | -8128.5 | 76.8(4) | < .001 | 16275 |
| *Explicit timing task (b)* | | | | |
| Interval duration | -8262.7 | | | 16531 |
| Interval duration × Age | -8220.8 | 83.7(2) | < .001 | 16452 |
| Interval duration × MMSE × Age | -8128.5 | 184.7(4) | < .001 | 16275 |
| *Implicit timing task (a)* | | | | |
| Interval duration | -2943.8 | | | 5896 |
| Interval duration × MMSE | -2921.7 | 44.11(2) | < .001 | 5855 |
| Interval duration × MMSE × Age | -2914.6 | 14.26(4) | .007 | 5849 |
| *Implicit timing task (b)* | | | | |
| Interval duration | 647.7 | | | 5896 |
| Interval duration × Age | -2937.2 | 13.24(2) | .001 | 5886 |
| Interval duration × MMSE × Age | -2914.6 | 45.13(4) | < .001 | 5849 |

with longer interval durations), it decreased with decreasing MMSE scores (Interval duration × MMSE interaction), but increased with greater age (Interval duration × Age interaction). The Interval duration × MMSE × Age three-way interaction was not significant.

## Discussion

In this correlational study, we compared explicit and implicit timing in both healthy and pathological older participants, thus, extending recent research on explicit and implicit timing in healthy older adults [2].

As concerns explicit timing, we reasoned that a slower clock in older participants should lead to a rightward shift of the psychometric curve with increasing age. By contrast, if changes in the temporal performance of older adults depend on their reduced cognitive control functions, one would expect a flatter psychometric curve with decreasing MMSE scores. These predictions were partly supported by our results, given that a flatter psychometric curve was observed not only for

**Table 2. Summary of the model outputs for the explicit timing task.**

| Fixed Effects | Odds ratios | β | p |
|---|---|---|---|
| (Intercept) | 1.082 | 0.103 | 0.093 |
| Interval duration | 1.842 | 0.665 | **<0.001** |
| MMSE | 1.034 | 0.046 | 0.521 |
| Age | 0.987 | -0.031 | 0.793 |
| Interval duration × MMSE | 1.245 | 0.149 | **<0.001** |
| Interval duration × Age | 0.877 | -0.099 | **<0.001** |
| MMSE × Age | 0.923 | -0.050 | 0.121 |
| Interval duration × MMSE × Age | 1.013 | 0.007 | 0.430 |
| $N_{ID}$ | | | 85 |
| Observations | | | 14729 |
| Marginal $R^2$ /Conditional $R^2$ | | | 0.30 / 0.38 |

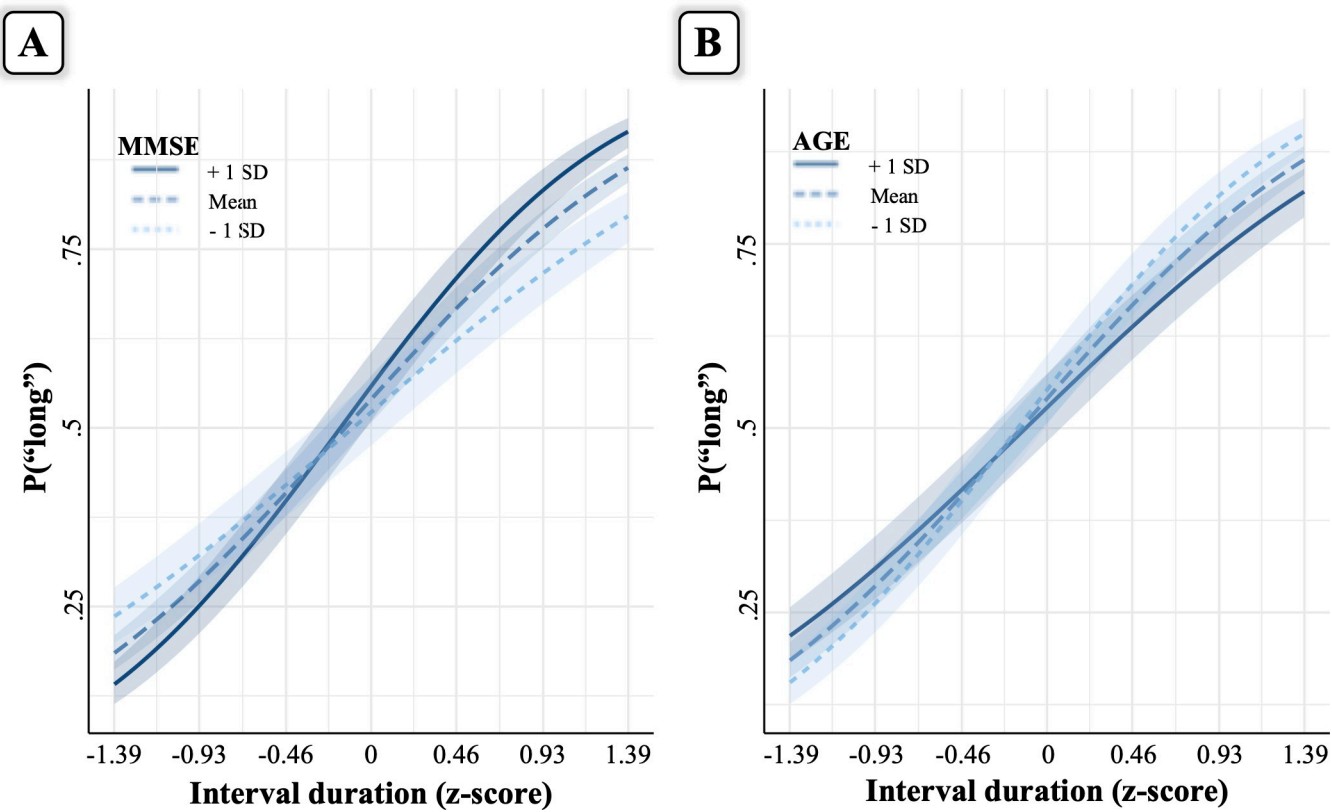

**Fig 3. Interaction effects in the explicit timing task.** Panel A depicts the interaction between Interval duration and MMSE, whereas panel B depicts the interaction between Interval duration and Age. The interaction plots were obtained using the "interact_plot" function of the R package interactions, which by default plots the marginal effects of the first continuous predictor (i.e., interval duration) at 1 standard deviation above (+1SD) and below (-1SD) the mean and at the mean itself of the second predictor (i.e., MMSE and Age, respectively). The seven interval durations on the x-axis represent the actual durations used in the task.

decreased MMSE scores but also for greater age. Therefore, much older and more compromised participants made less precise temporal judgments in the time bisection task.

The lack of evidence for a systematic over- or under-estimation bias in the time bisection task, indicated by the non-significant main effects of MMSE and age variables, accords well with

**Table 3. Summary of the model outputs for the implicit timing task.**

| Fixed Effects | Estimates | β | p |
|---|---|---|---|
| (Intercept) | 6.119 | -0.002 | < .001 |
| Interval duration | -0.064 | -0.166 | < .001 |
| MMSE | -0.161 | -0.328 | < .001 |
| Age | 0.118 | 0.279 | .001 |
| Interval duration × MMSE | -0.014 | -0.027 | < .001 |
| Interval duration × Age | -0.006 | -0.011 | .018 |
| MMSE × Age | 0.017 | 0.029 | .632 |
| Interval duration × MMSE × Age | 0.002 | 0.003 | .507 |
| N $_{ID}$ | | | 85 |
| Observations | | | 14976 |
| Marginal R$^2$ /Conditional R$^2$ | | | 0.18/0.66 |

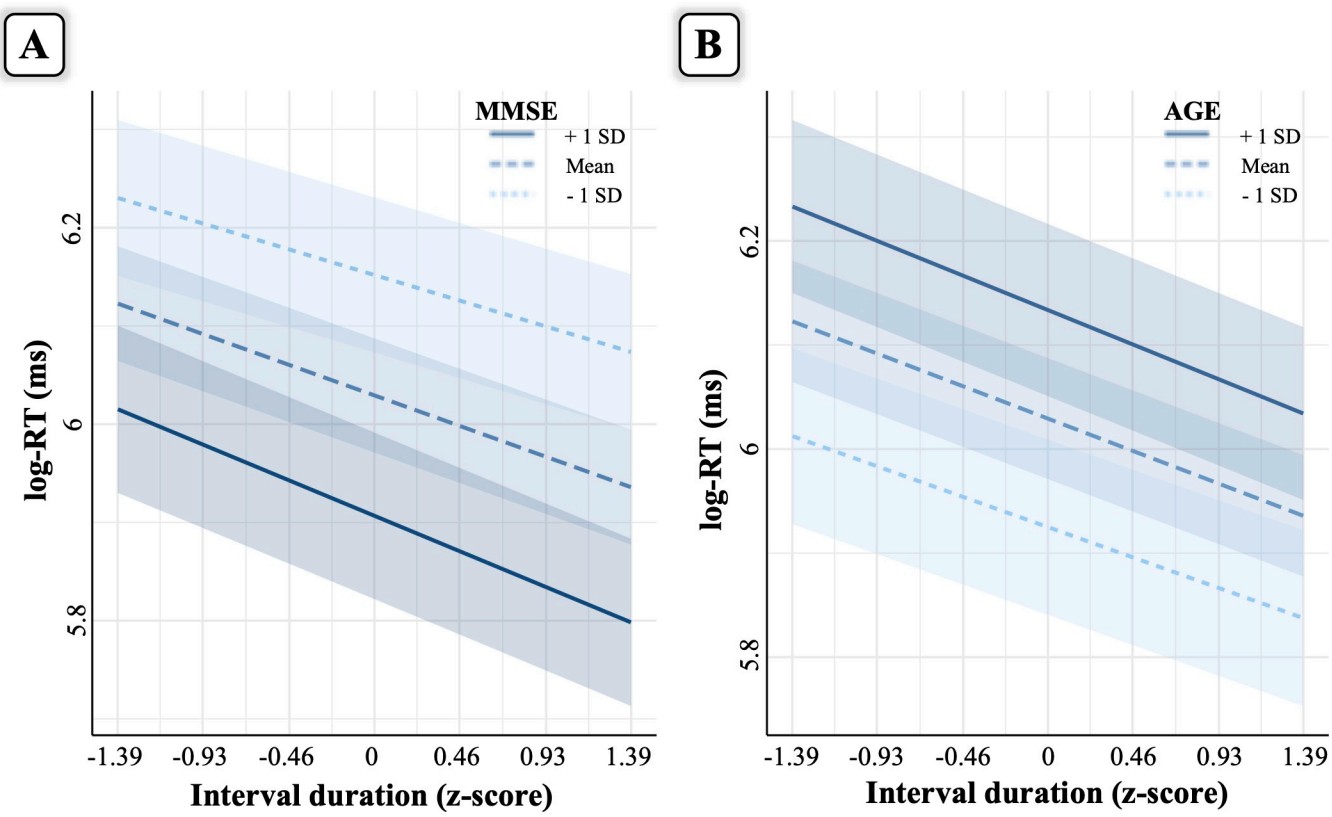

**Fig 4. Interaction effects in the implicit timing task.** Panel **A** depicts the interaction between Interval duration and MMSE, whereas panel **B** depicts the interaction between Interval duration and Age. The seven interval durations on the x-axis represent the actual durations used in the task. Please refer to Fig 3 for a detailed explanation of the interaction plots.

previous time bisection studies in healthy [3, 31, 32] and pathological samples [33, 34]. Consistent with this prior literature is also our observation of a flattening of the psychometric curve with a decreasing of MMSE scores and an increasing of age. A possible interpretation for the flatter psychometric curve associated with decreased MMSE scores or increased age is that more compromised or much older participants had a noisier memory representation of standard durations, which led them to respond "short" to long durations and "long" to short durations. Supporting this claim, it has been shown that similar anatomical structures underlie memory and timing functions [35]. Moreover, as stated in the Introduction, a role for memory in the performance of the time bisection task is acknowledged within pacemaker-based accounts of time perception [36–39]. Framing our results within such models, a poor memory representation of the standards would lead to a flatter psychometric curve. However, because we did not test for memory abilities, this explanation remains speculative and warrants further examination (but see [2] for relationships between performance on explicit timing and neuropsychological tests).

Overall, findings from the explicit time bisection task point to a deficit for much older or more compromised participants at the level of cognitive control functions rather than of temporal processing, as evinced by the flattening of the psychometric curve with increasing age and decreasing MMSE scores and the lack of a significant under- or over-estimation of interval durations. Future studies should administer older adults with tasks requiring judgment of different magnitudes (e.g., time, weight, brightness) to directly compare deficits in temporal and non-temporal dimensions.

In contrast to the explicit timing task, in the implicit timing one the task goal itself was non-temporal; participants had to simply respond to the onset of the cross without memorizing or providing an explicit judgment of the interval duration (or foreperiod) separating the cross from the thicker circle (see Fig 2A). The foreperiod effect (i.e., shorter RTs at longer interval durations), commonly observed in this type of RT tasks, is considered indicative of an implicit processing of time, with participants benefitting from the information afforded by the elapse of time during longer foreperiod trials [7, 20]. For the type of instructions given to participants and the task goal itself, implicit timing is believed to pose fewer demands on cognitive processes as compared to explicit timing. Accordingly, our original hypothesis was that the best predictor of performance on the implicit timing task should be the participants' age [2]. Echoing the results from the explicit timing task, the best fitting model was again the one including interval duration, MMSE scores and age variables. However, unlike the explicit timing task, MMSE scores and age had different effects on participants' performance. We detail these differences in what follows.

To begin with, the significant main effects of MMSE scores and age showed longer RTs for decreasing MMSE scores and greater age, in line with the presence of a generalized RT slowing in healthy and pathological older adults [40]. As alluded to above, this pattern does not reflect a deficit in the implicit processing of temporal information, which was instead operationalized by the size of the foreperiod effect. A close look at Fig 4 shows that the foreperiod effect increased with age, whereas it decreased with MMSE scores.

The presence of a larger foreperiod effect in much older participants is explained by the fact that they were slower than less older participants at shorter interval durations having, in turn, more room for improvement at longer durations. This is a simple and parsimonious explanation for the steepening of the foreperiod effect with increasing age. At the same time, a steeper foreperiod effect in much older participants indicates that the ability to implicitly process temporal information seems preserved with age, in line with previous studies [2, 41, 42]; but also see [43]). By contrast, looking at the MMSE score variable, although a decrease in the MMSE scores was associated with a slowing down of RT (as observed with increasing in Age), there was also a reduced foreperiod effect with decreased MMSE scores (differently from what observed for Age).

Taken together, our findings from the implicit timing task, showing that implicit processing of time is less efficient in participants with more cognitive decline, extend previous research in healthy aging [2] and add to other dissociations between explicit and implicit timing reported in children [44] and clinical conditions such as Parkinson's disease [45] and schizophrenia [46]. Coupled with these studies, our findings converge to the idea that implicit timing tasks could offer important insights into processing of time in vulnerable populations because of its fewer demands in terms of cognitive control functions (i.e., no explicit instructions to pay attention to time neither to memorize durations) as compared to more demanding explicit timing tasks, where deficits in temporal processing might be masked by other executive deficits. A path that future research must surely follow is the investigation of the neural mechanisms that are in charge of preserving implicit timing in healthy aged populations. It would be also interesting to find the age point at which performance on explicit and implicit timing tasks starts to diverge to get a comprehensive picture on processing of explicit and implicit timing from a life-span perspective.

A potential caveat to our study is that we exclusively relied on the use of the MMSE and did not employ other neuropsychological tests that could have better defined the cognitive profile of our sample. Nonetheless, it should be noted that the MMSE is routinely used in clinical practice to screen for cognitive impairment because of its easy procedure and short administration time. Another concern could be that our study did not provide any information about

potential differences in temporal processing between individuals diagnosed with MCI and patients with dementia. As amply discussed above, we were interested in disentangling the effects of age from those related to cognitive decline regardless of the specific neurodegenerative condition. However, our study could be used in the future as an ideal starting point to investigate whether and how neurodegenerative conditions differently affect performance on temporal tasks.

In conclusion, the present findings contribute to the understanding of deficits in explicit and implicit timing tasks in older adults by showing dissociable associations with age and cognitive decline.

## Supporting information

**S1 Table. Descriptive statistics (mean and standard deviation) of our sample.**
(DOCX)

**S2 Table. Summary of the model outputs for the explicit timing task from analyses including participants with a proportion of non-given responses lower than .1, .2, .3, or .4, respectively.**
(DOCX)

**S3 Table. Summary of the model outputs for the implicit timing task from analyses including participants with a proportion of non-given responses lower than .1, .2, .3, or .4, respectively.**
(DOCX)

## Acknowledgments

This work was carried out within the scope of the project "Use-inspired basic research", for which the Department of General Psychology of the University of Padova has been recognized as "Dipartimento di Eccellenza" by the Ministry of University and Research. The authors gratefully thank Diletta Piazzesi, Martina Corda, Davide Tempesta and Alessandro Vicenzotti for helping with data collection.

## Author Contributions

**Conceptualization:** Mariagrazia Capizzi, Giovanna Mioni.

**Data curation:** Antonino Visalli, Alessio Faralli, Giovanna Mioni.

**Formal analysis:** Antonino Visalli.

**Investigation:** Giovanna Mioni.

**Methodology:** Mariagrazia Capizzi, Antonino Visalli, Giovanna Mioni.

**Resources:** Alessio Faralli, Giovanna Mioni.

**Software:** Giovanna Mioni.

**Supervision:** Mariagrazia Capizzi, Alessio Faralli, Giovanna Mioni.

**Writing – original draft:** Mariagrazia Capizzi, Antonino Visalli, Giovanna Mioni.

**Writing – review & editing:** Mariagrazia Capizzi, Antonino Visalli, Giovanna Mioni.

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
