## [Decision Letter · Decision Letter 0]

19 Oct 2021

PONE-D-21-29634Explicit and implicit timing in healthy and pathological aging:

Disentangling the role of age and cognitive functioning in temporal processingPLOS ONE

Dear Dr. Mioni,

Thank you for submitting your manuscript to PLOS ONE. After careful consideration, we feel that it has merit but does not fully meet PLOS ONE’s publication criteria as it currently stands. Therefore, we invite you to submit a revised version of the manuscript that addresses the points raised during the review process.

We look forward to receiving your revised manuscript.

Kind regards,

Bradley R. King

Academic Editor

PLOS ONE

Journal Requirements:

Reviewers' comments:

Reviewer's Responses to Questions

**Comments to the Author**

1. Is the manuscript technically sound, and do the data support the conclusions?

Reviewer #1: Yes

Reviewer #2: Partly

2. Has the statistical analysis been performed appropriately and rigorously? 

Reviewer #1: Yes

Reviewer #2: I Don't Know

3. Have the authors made all data underlying the findings in their manuscript fully available?

Reviewer #1: Yes

Reviewer #2: No

4. Is the manuscript presented in an intelligible fashion and written in standard English?

Reviewer #1: Yes

Reviewer #2: Yes

5. Review Comments to the Author

Reviewer #1: This is a relatively straightforward study of timing in older participants whose MMSE scores were correlated with performance on both explicit (temporal bisection) and implicit (variable foreperiod) measures of timing. The authors conclude that explicit timing is impaired due to cognitive dysfunction but implicit timing is spared.

Major comments

1. In the Methods, authors state that they included participants with Alzheimer’s Disease (AD) and Mild Cognitive Impairment (MCI) as well as healthy older adults. Table 1 indicates that MMSE scores were used as cut-offs for healthy/MCI/dementia groups. However, it’s not clear whether participants had received a formal diagnosis of MCI or AD, or whether they were classified as AD or MCI purely on the basis of their MMSE scores. Please clarify whether the “pathological” aspect of the study was based on formal diagnosis, or on the MMSE scores obtained by the researchers. This is clarification is important for two reasons (1) the inclusion of pathological groups is the only thing that differentiates this study from previously published findings so precision concerning diagnoses is critical (2) dementia is not the same thing as AD. It’s possible to have low MMSE scores for reasons other than AD. So we need to know whether participants were classified as having dementia based on MMSE scores or whether there were a group of patients that had received a diagnosis of AD. If it’s the former, then all mention of participants with AD is misleading and should be removed from the manuscript.

2. Given the schematics presented in Figures 1B and 2B, the authors should analyse their explicit timing data in terms of the bisection point and slope of the bisection curve, and their implicit timing data in terms of the slope of the RT function. These measures directly index the various predictions detailed in the figures. They can be calculated on a participant-by participant level and will provide a more accurate reflection of individual differences, and how they are modulated by MMSE score or age.

3. At the end of the Introduction, the authors state that “A more direct way to disentangle the contribution of age and cognitive resources to processing of explicit and implicit timing is to include both healthy and pathological aged samples and to use participants’ age and level of cognitive decline as continuous predictors of performance.” I don’t see why this is more “direct” than the Droit-Volet study. In the current study, they using a correlational approach with age and cognitive scores as a parametric variables, which is exactly what Droit-Volet did. The only difference is that they use MMSE as a measure of cognitive function, rather than neuropsychological tests of attention and memory. The authors need to clarify why they think their approach is more “direct”.

4. A few issues concerning the novelty of the study:

- In the Abstract the authors claim to provide “the first experimental evidence that processing of implicit, but not explicit, timing is differentially affected in healthy and pathological aging”. This is a subtle point but potentially misleading – their investigation might be the first demonstration of differential effects of DEMENTIA on implicit and explicit timing but it’s not the first demonstration of this in healthy aging. This sentence should be reformulated to reflect the novel contribution of the study, which is the inclusion of participants with MMSE scores that reflect MCI or mild to moderate dementia.

- Their predictions at the end of the Intro are exactly the same as the results of the Droit-Volet study. The authors need to clarify the novel contribution of their own study, perhaps by reframing the predictions in terms of pathological decline and the four levels of dementia detailed in Table 1?

- The authors conclude that they provide the “first evidence that implicit processing of time can be spared in healthy age-related cognitive decline”. But this is NOT the first evidence of such a finding. They mention in the manuscript several studies that have already shown a steeper FP effect in older participants and in the Droit-Volet study implicit timing performance in older participants was influenced more by the hazard function than by a memorized representation of duration, again suggesting sparing of implicit timing. In addition, they conclude that explicit timing was impaired due to general cognitive dysfunction, but implicit timing was spared. This is exactly the conclusion reached by Droit-Volet et al, so it’s unclear what the novel contribution of this study is. The authors also say that it would be interesting to apply their approach to neuropsychological tests rather than the MMSE, but this is precisely what was done in the Droit-Volet study! The similarity of their findings and conclusions to those of previous studies must be mentioned and the novel contribution of the current study needs to be radically clarified. In addition, all statements to the effect of providing the “first evidence” are vastly overstated and should be removed.

- The authors find an effect of both age and MMSE on explicit timing. This was also the case in the Droit-Volet study, who addressed this issue with a hierarchical regression analysis to determine whether age or cognitive function score was the best predictor of performance. They found that age was no longer a significant predictor once attention was modelled. In the discussion the authors speculate that poor memory or poor attention could explain the effects of MMSE and age on explicit timing, but Droit-Volet tested this with a hierarchical regression analysis and found attention to be the best predictor of performance. Indeed, once attention was included in the model age no longer predicted performance. These findings are relevant for the conclusions of the current study (which used a global measure of cognitive function (MMSE) rather than scores on individual cognitive processes) and should be mentioned in the discussion.

Minor comments

1. In the Introduction, I’m not sure that the description of results in terms of the pacemaker-accumulator model is strictly necessary. It doesn’t inform either the predictions or results. I suggest removing all this part from “Such psychological operations are typically explained…” to “(Lustig and Meck, 2001)”.

2. In Figures 3 and 4, the authors present data in terms of the mean, +1SD and -1 SD. But no explanation of these categories is provided. Why was this approach chosen?

3. In the discussion the authors state that figure 4 “clearly shows” that the FP effect increased with age (and decreased with MMSE). Based on their statement I would expect a flatter slope for the -1SD MMSE group and a steeper slope for the +1SD age group. The figure does “clearly” show the main effect of age and the main effect of MMSE (upward or downward shifts in the slopes) but it doesn’t “clearly” show the strength of the FP effect, which would be indexed by a change in the SLOPE of the RT function. The slopes for the three categories all look parallel. I don’t question their statistics but I wonder if the can authors find a more descriptive/illustrative way to present the data?

Reviewer #2: The authors conducted a behavioral study on temporal processing in older adults with varying degrees of cognitive performance as assessed on the MMSE. Using two dissimilar paradigms, they distinguished explicit and implicit timing abilities, which might be differentially affected by aging, age-related changes in cognition, or both. This is a very interesting research question, and it is very informative to advance the current knowledge of differences between different aspects of timing abilities (explicit vs. implicit) and how these abilities relate to interindividual differences in cognitive status. I do have some conceptual concerns, though, and hope that my comments will help to address these.

My main concern is how cognitive functions are regarded in this paper. First, it is somehow implied that there is some kind of a mechanistic role for cognitive functions in temporal processing. This is never made very explicit, but it should be clarified that by relying on interindividual differences, it is impossible to draw conclusions regarding the mechanistic involvement of cognitive functions in temporal processing (see the paper by Borsboom and colleagues (2009) for a more detailed account, https://dx.doi.org/10.1007/978-0-387-95922-1_4). Second, the research design seems to imply a distinction between cognitive functions and temporal processing that should be subjected to critical conceptual discussion. Could it be argued that temporal processing is an aspect of cognitive functions, rather than a consequence? Is there any evidence that these are distinguishable entities at all? If not, how could one possibly “influence” the other (but see my previous comment)? Third, the assessment of cognitive functions using the MMSE screening as a single measure does not seem broad enough to draw firm conclusions. The authors address this in the limitations already, but the conclusions drawn from the study do not reflect this limitation. I would strongly suggest to temper all claims regarding “cognitive functions” such that it becomes clear at all times that this refers to a rather superficial cognitive screening, rather than an in-depth or in any way specific assessment of cognitive functioning.

In addition to that, the wording sometimes obscures the exact comparisons that were being made in the paper. Sometimes, I had the impression that the authors aimed at comparing timing abilities between two dissimilar groups (healthy vs. “pathological” aging), but the analyses refer to the entire sample. I believe that this is preferable, given that group comparisons would not be powerful enough, but the wording should be adapted accordingly, also in the Manuscript Title. In addition, the word “aging” seems inappropriate here, as there was no longitudinal or group comparison, but this was instead a cross-sectional study in a population of older adults. This should be made very clear throughout the text.

More detailed comments:

Abstract:

The wording in the results and discussion sections of the abstract should be adapted for clarity. For instance, “… processing of implicit, but not explicit, timing is differentially affected in healthy and pathological aging” seems to suggest that patterns of timing abilities significantly differ between healthy and non-healthy older adults.

Introduction:

The concept of “temporal processing” should be introduced in more detail. What does it entail, and from what related concepts does it differ? How is temporal processing distinguished from cognitive functioning in general, if at all (see above)?

Paragraph starting with “To our knowledge, … “: The second sentence referring to previous studies is somewhat ambiguous and should be revised.

Same paragraph, last portion: It is not valid to conclude that lower attentional abilities are in any way mechanistically related to temporal processing, please see the paper by Borsboom that I recommended above. This must be revised.

Same paragraph, last sentence + first sentence following paragraph: Is this a statistically meaningful difference? Is it possible that cognitive scores were just not sensible enough to show an effect, e.g. due to the typically low reliability of cognitive measures?

Paragraph starting with “Collectively, …”, third sentence: The current design is implied to provide more direct evidence for the effect of interest than previous studies, but this is not the case, as it relies on the same correlational logic, rather than on a true experimental manipulation of “age” and “cognition”. This is not a problem per se (or at least not specific to this particular paper), but it should be discussed accordingly.

Paragraph starting with “Different, …”, third sentence: Remove the comma before “which”, as it changes the meaning of the sentence.

Same paragraph, last sentence: It seems very difficult to show that two variables are “unrelated”, as this implies acceptance of the null hypothesis. Please revise the wording.

Methods:

As a general suggestion, I would recommend to walk the reader more gently through the analyses and explain which particular questions are answered by which tests and what is implied by which outcome.

General comment: MMSE-scores are not independent of participant age. Does this represent a problem for the chosen analysis approach? Consider adapting the analysis strategy, but at least, this issue should be discussed.

Regarding the strategy to deal with no-response trials: Did I understand correctly that no-response trials were included in the original analyses? Why were they not discarded on a trial-by-trial basis in that analysis, rather than discarding an entire case in a supplementary analysis? Would the results change when excluding no-response trials?

Results:

Paragraph “Implicit timing task”, second sentence: Could you please explain in more detail what it means that “the model was a bit stressed”?

Same paragraph, last sentence: This is very difficult to see from the Figure. Please provide some more guidance or adapt accordingly.

Discussion:

First paragraph: The statement “in both healthy and pathological aged populations” is misleading; cf. my earlier comment. Please revise.

Second paragraph, last sentence: The term “pathological” cognitive decline is not appropriate here, given that (1) the classification as “pathology” was merely based on a screening tool rather than on a solid clinical diagnosis, and (2) it was not part of an actual comparison. Please rephrase.

Third paragraph, sentence starting with “Another possible, …”: It is suggested here that attentional deficits might explain worse performance in relatively older adults and in relative low performers on the MMSE on the explicit timing task. Such a deficit is not definitely shown, as there was no specific test to measure attention. Even if an attention deficit had been shown to coincide with worse explicit timing performance, it should be made explicit that this does in no way allow for a mechanistic or causal interpretation of the results. Finally, if attention were indeed causally related to timing abilities, why would this effect be limited to the explicit timing task and not also the implicit timing task?

Same paragraph, sentence starting with “In any case, …”: This wording suggests that age and cognitive performance independently modulate task performance, but they are confounded. Please rephrase, if applicable (cf. my earlier comment regarding the Methods).

Same paragraph, sentence starting with “Remarkably, …”: This sentence should be rephrased. First, it is difficult to conclude that “neither age nor cognitive impairment altered timekeeping mechanisms”, as this implies the acceptance of the null hypothesis. Second, it should not be stated that either of those variables “influenced” cognitive processes to avoid suggesting a mechanistic relationship.

Fourth paragraph, sentence starting with “Considering that implicit timing poses…”: Can it actually be stated that implicit timing poses less demands on cognitive processing as compared to explicit timing? What is the basis for this claim?

Paragraph starting with “The presence of a larger…”, last sentence: Revise the wording in this sentence. It seems to suggest a comparison between “older”/ “healthy” and “impaired” / “pathological” adults (cf. my previous comments).

Paragraph starting with “Taken together, …”, third sentence: Please remove the statement regarding early markers. This cross-sectional design does not allow for the conclusion of any longitudinal effects. In addition, I understood that the authors conceptualized temporal processing to be distinct from cognitive functioning. How can this be reconciled with the viewpoint that temporal processing can serve as a marker for cognitive changes?

Same paragraph, fourth sentence: If this paradigm poses “very little demands” on cognitive resources, then why is it worth to assess a relationship with MMSE scores at all?

Same paragraph, fifth sentence: This wording implies a direct statistical comparison between explicit and implicit timing tasks, please revise. Similarly, a few sentences further, it is stated that the MMSE was “enough sensitive to differentiate performance on explicit and implicit tasks”, but this, again, would need to be shown statistically.

Last paragraph, second sentence: It seems difficult to make conclusions regarding “age-related” changes based on the present data, as the study population were older adults exclusively.

6. PLOS authors have the option to publish the peer review history of their article (what does this mean?). If published, this will include your full peer review and any attached files.

Reviewer #1: No

Reviewer #2: No

---

## [Author Response · Author response to Decision Letter 0]

6 Jan 2022

Reviewers' comments:

Reviewer's Responses to Questions

Comments to the Author

1. Is the manuscript technically sound, and do the data support the conclusions?

Reviewer #1: Yes

Reviewer #2: Partly

2. Has the statistical analysis been performed appropriately and rigorously?

Reviewer #1: Yes

Reviewer #2: I Don't Know

3. Have the authors made all data underlying the findings in their manuscript fully available?

Reviewer #1: Yes

Reviewer #2: No

4. Is the manuscript presented in an intelligible fashion and written in standard English?

Reviewer #1: Yes

Reviewer #2: Yes

5. Review Comments to the Author

Reviewer #1: 

This is a relatively straightforward study of timing in older participants whose MMSE scores were correlated with performance on both explicit (temporal bisection) and implicit (variable foreperiod) measures of timing. The authors conclude that explicit timing is impaired due to cognitive dysfunction but implicit timing is spared.

RESPONSE: We thank the reviewer for the positive evaluation of our work.

Major comments

1. In the Methods, authors state that they included participants with Alzheimer’s Disease (AD) and Mild Cognitive Impairment (MCI) as well as healthy older adults. Table 1 indicates that MMSE scores were used as cut-offs for healthy/MCI/dementia groups. However, it’s not clear whether participants had received a formal diagnosis of MCI or AD, or whether they were classified as AD or MCI purely on the basis of their MMSE scores. Please clarify whether the “pathological” aspect of the study was based on formal diagnosis, or on the MMSE scores obtained by the researchers. This is clarification is important for two reasons (1) the inclusion of pathological groups is the only thing that differentiates this study from previously published findings so precision concerning diagnoses is critical (2) dementia is not the same thing as AD. It’s possible to have low MMSE scores for reasons other than AD. So we need to know whether participants were classified as having dementia based on MMSE scores or whether there were a group of patients that had received a diagnosis of AD. If it’s the former, then all mention of participants with AD is misleading and should be removed from the manuscript.

RESPONSE: We apologize for the lack of clarity concerning these issues. Because our study was not aimed at investigating whether and how different neurodegenerative conditions affect performance on explicit and implicit timing tasks (as also outlined in the Introduction and Discussion Sections), any reference to specific types of dementia (e.g., Alzheimer’s disease) was deleted from the manuscript. As specified in the revised version of the manuscript (see pages 8 and 9), unhealthy participants received a formal diagnosis of MCI or dementia; accordingly, they all can be confidently considered as pathological. Nevertheless, although participants received such a diagnosis, any reference to cognitive impairment/decline in our sample refers to the MMSE score, which, for our goals, was used as a continuous predictor of performance. This aspect is now clearly reported on page 8 (“although unhealthy participants received a formal diagnosis of (mild-) cognitive dementia (see the Methods Section), hereafter any reference to cognitive impairment/decline in our sample is related to the MMSE score. Participants’ ages and MMSE scores were considered as continuous predictors of performance in the analyses”). 

Finally, to avoid confusion regarding this part, namely, that no cut-offs were used to characterize our sample, we amended the legend of the Supplementary Table S1, which now reads as follows: “According to the cut-offs commonly used in the literature, a score between 30 and 28 would define healthy older adults with a normal cognitive functioning; a score between 27 and 25 would indicate the presence of Mild Cognitive Impairment (MCI); a score between 24 and 19 would indicate a mild dementia, whereas a score between 18 and 10 a moderate dementia”. Moreover, we specified on page 9 that “for completeness Supplementary table S1 also reports the clinical classifications (i.e., dementia or MCI) of our sample as commonly done according to the cut-offs used in the literature”.

2. Given the schematics presented in Figures 1B and 2B, the authors should analyse their explicit timing data in terms of the bisection point and slope of the bisection curve, and their implicit timing data in terms of the slope of the RT function. These measures directly index the various predictions detailed in the figures. They can be calculated on a participant-by participant level and will provide a more accurate reflection of individual differences, and how they are modulated by MMSE score or age.

RESPONSE: We thank the reviewer for the suggestion. However, if we get it correctly, the reviewer proposes to perform a two-level regression, which is a form of multilevel modelling analysis. Namely, we should fit logistic (in the explicit timing task) and linear (in the implicit timing task) regressions separately for each participant (first-level subject-specific analysis), and then use the obtained parameters for fitting models at a second (group) level including age and MMSE scores. We agree that the proposed approach is a way to test the hypotheses presented in Figures 1B and 2B. However, we would respectfully point out that this approach is already implemented in our mixed effects analyses. Our models, indeed, are another form of multilevel modelling in which the suggested first and second levels analyses are fitted simultaneously. Accordingly, and unless the reviewer meant something else, we have already implemented the suggested approach with our mixed effect analyses. Please also note that, in response to Reviewer 2, we added a more thorough explanation of the performed analyses and interpretation of the results on pages 11 and 12. 

3. At the end of the Introduction, the authors state that “A more direct way to disentangle the contribution of age and cognitive resources to processing of explicit and implicit timing is to include both healthy and pathological aged samples and to use participants’ age and level of cognitive decline as continuous predictors of performance.” I don’t see why this is more “direct” than the Droit-Volet study. In the current study, they using a correlational approach with age and cognitive scores as a parametric variables, which is exactly what Droit-Volet did. The only difference is that they use MMSE as a measure of cognitive function, rather than neuropsychological tests of attention and memory. The authors need to clarify why they think their approach is more “direct”.

RESPONSE: We agree with the reviewer that our statement was unclear, and we deleted the term “direct” from the Introduction. Indeed, what we wanted to emphasize here is that our study extended previous investigation on processing of explicit and implicit timing in older adults by including both healthy and pathological older participants. In the light of this comment, we rephrased the paragraph as follows (see page 8): “In the present study, we aimed to advance our knowledge about performance of older adults on explicit and implicit timing tasks by considering not only healthy but also pathological older participants. This allowed capturing differences in explicit and implicit timing tasks linked to age and pathological cognitive decline, two variables that, although often correlated, are not systematically associated (Jansen et al., 2018). To this end, healthy older adults and individuals diagnosed with either Mild Cognitive Impairment (MCI) or dementia completed explicit (time bisection) and implicit (foreperiod) timing tasks in a single session. The Mini-Mental State Examination (MMSE; Folstein, Folstein, & McHugh, 1975) was used as an index of cognitive decline. The MMSE represents, indeed, one of the most routinely used screening tools in clinical practice, even if it provides only a generic assessment of cognitive decline. Of note, although unhealthy participants received a formal diagnosis of (mild-) cognitive dementia (see the Methods Section), hereafter any reference to cognitive impairment/decline in our sample is related to the MMSE score”. 

Moreover, we converge with the reviewer on the similarity between the two correlational approaches. However, we respectfully disagree with the reviewer in that our sample size (N= 85) is more adequate for testing correlations than the one used by Droit-Volet et al. (N=20). We clarified this point in the manuscript (page 9): “this sample allows observing a correlation with a Pearson’s r of .3 with a power of .8”. 

4. A few issues concerning the novelty of the study:

4.1- In the Abstract the authors claim to provide “the first experimental evidence that processing of implicit, but not explicit, timing is differentially affected in healthy and pathological aging”. This is a subtle point but potentially misleading – their investigation might be the first demonstration of differential effects of DEMENTIA on implicit and explicit timing but it’s not the first demonstration of this in healthy aging. This sentence should be reformulated to reflect the novel contribution of the study, which is the inclusion of participants with MMSE scores that reflect MCI or mild to moderate dementia.

RESPONSE: The Abstract has been amended to incorporate the concerns raised by both reviewers. Moreover, any reference to “first evidence” has been deleted from both the Abstract and the manuscript. We think that the novel contribution of our study is overall clearer. The last paragraph of the Abstract now reads as follows: “Results for the explicit timing task showed a flatter psychometric curve with increasing age or decreasing MMSE scores, pointing to a deficit at the level of cognitive control functions rather than of temporal processing. By contrast, for the implicit timing task, a decrease in the MMSE scores was associated with a reduced foreperiod effect, an index of implicit time processing. Overall, these findings extend previous studies on explicit and implicit timing in healthy aged samples by dissociating between age and cognitive decline (in the normal-to-pathological continuum) in older adults

4.2- Their predictions at the end of the Intro are exactly the same as the results of the Droit-Volet study. The authors need to clarify the novel contribution of their own study, perhaps by reframing the predictions in terms of pathological decline and the four levels of dementia detailed in Table 1?

RESPONSE: Following this point, we clarified on page 8 that: “Concerning implicit timing, we expected to find a significant association of performance on the foreperiod task with age rather than with MMSE scores, replicating previous research (Droit-Volet et al., 2019)”. 

By contrast, our predictions for the explicit timing task were not exactly the same as the results of the Droit-Volet et al.’ study. Indeed, such a previous study found no differences in accuracy performance between younger and older participants. Here, we hypothesized (please also see Figure 1B) that: “As concerns explicit timing, if the poor performance of older participants on the time bisection task depends on a deficit in temporal processing, this should be reflected by a (rightward) shift of the psychometric curve. According to previous literature reporting a slowing down of the pacemaker with age (see Turgeon, Lustig, & Meck, 2016), we predicted to find a significant association between the rightward shift of the curve and age. If, conversely, changes in the performance of older participants depend on their reduced cognitive control functions, this should translate into a flatter psychometric curve. Therefore, we predicted a significant association between MMSE scores and the flattening of the curve”. Accordingly, both scenarios (and not only the second one) are equally plausible. Please also note that, as reported in the Discussion section (pages 20-21), we were not interested in the differences between individuals diagnosed with MCI or dementia, but in considering the MMSE scores as a continuous predictor of performance. In this sense, we do not have any specific prediction as a function of the classification reported in Table 1, which was only inserted for the sake of completeness and for transparently providing the reader with all the information about our sample (see also our response to point 1). 

4.3- The authors conclude that they provide the “first evidence that implicit processing of time can be spared in healthy age-related cognitive decline”. But this is NOT the first evidence of such a finding. They mention in the manuscript several studies that have already shown a steeper FP effect in older participants and in the Droit-Volet study implicit timing performance in older participants was influenced more by the hazard function than by a memorized representation of duration, again suggesting sparing of implicit timing. In addition, they conclude that explicit timing was impaired due to general cognitive dysfunction, but implicit timing was spared. This is exactly the conclusion reached by Droit-Volet et al, so it’s unclear what the novel contribution of this study is. The authors also say that it would be interesting to apply their approach to neuropsychological tests rather than the MMSE, but this is precisely what was done in the Droit-Volet study! The similarity of their findings and conclusions to those of previous studies must be mentioned and the novel contribution of the current study needs to be radically clarified. In addition, all statements to the effect of providing the “first evidence” are vastly overstated and should be removed.

RESPONSE: As reported in our response to point 4.1, all the references to “first evidence” were removed. Moreover, the paragraph concerning the administration of neuropsychological tasks in future studies was also removed. The novel contribution of the present study concerning the finding of a reduced foreperiod effect in more compromised participants was clarified on page 20: “by contrast, looking at the MMSE score variable, although a decrease in the MMSE scores was associated with a slowing down of RT (as observed with increasing in Age), there was also a reduced foreperiod effect with decreased MMSE scores (differently from what observed for Age)”.

4.4- The authors find an effect of both age and MMSE on explicit timing. This was also the case in the Droit-Volet study, who addressed this issue with a hierarchical regression analysis to determine whether age or cognitive function score was the best predictor of performance. They found that age was no longer a significant predictor once attention was modelled. In the discussion the authors speculate that poor memory or poor attention could explain the effects of MMSE and age on explicit timing, but Droit-Volet tested this with a hierarchical regression analysis and found attention to be the best predictor of performance. Indeed, once attention was included in the model age no longer predicted performance. These findings are relevant for the conclusions of the current study (which used a global measure of cognitive function (MMSE) rather than scores on individual cognitive processes) and should be mentioned in the discussion.

RESPONSE: As correctly pointed out by Reviewer 2, since we did not measure attentional abilities, all the speculations concerning the involvement of attention were removed from the manuscript. Moreover, as concerns the role of memory, we now specified (page 18) that “because we did not test for memory abilities, this explanation remains speculative and warrants further examination (but see Droit-Volet et al., 2019, for relationships between performance on explicit timing and neuropsychological tests)”.

Minor comments

1. In the Introduction, I’m not sure that the description of results in terms of the pacemaker-accumulator model is strictly necessary. It doesn’t inform either the predictions or results. I suggest removing all this part from “Such psychological operations are typically explained…” to “(Lustig and Meck, 2001)”.

RESPONSE: As reported in our response to Reviewer 2 (see point 2), the description of pacemaker-based model of time is necessary for framing the concepts of “temporal processing” and “cognitive control functions” and for better understanding the predictions illustrated in Figures 1B and 2B. We, thus, hope that the new elaboration regarding this aspect (page 3) is more informative as compared to the previous version. 

2. In Figures 3 and 4, the authors present data in terms of the mean, +1SD and -1 SD. But no explanation of these categories is provided. Why was this approach chosen?

RESPONSE: We apologize for the poor explanation of the Figures. This approach is a simple and common solution to represent interactions between two continuous predictors (e.g., interval duration and age; concerning the commonality, it is the default behavior of the function interact_plot in the R package "interactions"). A real alternative would be to plot a 3D plane that would be hard to visualize (unless an interactive visualization mode is available).

The aspect of the default behavior of the function has now been explicated in the figure caption (page 15) as follows: “Figure 3. Interaction effects in the explicit timing task. Panel A depicts the interaction between Interval duration and MMSE, whereas panel B depicts the interaction between Interval duration and Age. The interaction plots were obtained using the "interact_plot" function of the R package interactions, which by default plots the marginal effects of the first continuous predictor (i.e., interval duration) at 1 standard deviation above (+1SD) and below (-1SD) the mean and at the mean itself of the second predictor (i.e., MMSE and Age, respectively). The seven interval durations on the x-axis represent the actual durations used in the task.”

3. In the discussion the authors state that figure 4 “clearly shows” that the FP effect increased with age (and decreased with MMSE). Based on their statement I would expect a flatter slope for the -1SD MMSE group and a steeper slope for the +1SD age group. The figure does “clearly” show the main effect of age and the main effect of MMSE (upward or downward shifts in the slopes) but it doesn’t “clearly” show the strength of the FP effect, which would be indexed by a change in the SLOPE of the RT function. The slopes for the three categories all look parallel. I don’t question their statistics but I wonder if the can authors find a more descriptive/illustrative way to present the data?

RESPONSE: We understand that the figure doesn’t “clearly” show the strength of the FP effect in a very intuitive way. However, even if the slopes for the three categories all look parallel, they are not parallel. Otherwise, if the slopes were equal (parallel lines), the interaction between interval duration and age and the interaction between interval duration and MMSE would not have been significant. Said this, given that changes in slope might not be so “clearly” evident in Figure 4, we amended the sentence as follows (page 19): “A close look at Figure 4 shows that the foreperiod effect increased with age, whereas it decreased with MMSE scores.” Unfortunately, indeed, there are no other illustrative ways to present the data as have been analyzed here. We, thus, hope that our amendment is sufficient to meet the reviewer's request. 

Reviewer #2: 

The authors conducted a behavioral study on temporal processing in older adults with varying degrees of cognitive performance as assessed on the MMSE. Using two dissimilar paradigms, they distinguished explicit and implicit timing abilities, which might be differentially affected by aging, age-related changes in cognition, or both. This is a very interesting research question, and it is very informative to advance the current knowledge of differences between different aspects of timing abilities (explicit vs. implicit) and how these abilities relate to interindividual differences in cognitive status. I do have some conceptual concerns, though, and hope that my comments will help to address these.

RESPONSE: We thank the reviewer for the positive evaluation of our work and for the useful comments. 

My main concern is how cognitive functions are regarded in this paper. 

1-First, it is somehow implied that there is some kind of a mechanistic role for cognitive functions in temporal processing. This is never made very explicit, but it should be clarified that by relying on interindividual differences, it is impossible to draw conclusions regarding the mechanistic involvement of cognitive functions in temporal processing (see the paper by Borsboom and colleagues (2009) for a more detailed account, https://dx.doi.org/10.1007/978-0-387-95922-1_4). 

RESPONSE: We thank the reviewer for the useful comment and for the suggested reading. As also pointed out in our response to the below issue n. 2, we hope that our reference to “temporal processing” and “cognitive control functions” is now clearer in the manuscript, and that we managed to explain that our study was not aimed at drawing any conclusion regarding the involvement of cognitive functions in temporal processing. Specifically, as reported on page 3, in framing our study, we built up on pacemaker-based models of time to address the question of whether “the age-related changes observed in explicit timing tasks can be genuinely attributed to a dysfunction at the level of the clock stage, hereafter referred to as “temporal processing” (e.g., a slower clock accumulating more pulses in older than younger adults), or should be rather considered as a secondary deficit at the level of memory/decision stages, hereafter referred to as “cognitive control functions” (e.g., a noisier memory representation of the short and long standards in older adults)”. In other words, “temporal processing” refers to the clock stage, whereas “cognitive control functions” to memory/decision stages, which, in influential pacemaker-based models of time, represent the three stages accounting for explicit temporal judgements. 

Said this, we again apologize if it seemed that our goal was to account for the mechanisms involved in temporal processing and we hope that it is now clear that, by building up on pacemaker-based models of time, we rather aimed at exploring possible differences in older adults in explicit and implicit timing tasks. 

2-Second, the research design seems to imply a distinction between cognitive functions and temporal processing that should be subjected to critical conceptual discussion. Could it be argued that temporal processing is an aspect of cognitive functions, rather than a consequence? Is there any evidence that these are distinguishable entities at all? If not, how could one possibly “influence” the other (but see my previous comment)? 

RESPONSE: In keeping with our previous response, the distinction between “temporal processing” and “cognitive control functions” is grounded in pacemaker-based models of time. As also pointed out in the description of Figure1B, we acknowledge that “it is difficult to completely isolate clock (i.e., “temporal processing”) from memory/decision stages (i.e., “cognitive control functions”)”. This notwithstanding, it makes sense to hypothesise that “age-related changes in clock speed (namely, in temporal processing) should be mainly expressed by a rightward shift of the psychometric curve (i.e., a slower clock in older adults). By contrast, a flatter psychometric curve in older adults could be likely attributed to a deficit in the additional cognitive control functions (e.g., working memory) thought to be required to correctly perform on the time bisection task”. We hope that these clarifications allow for a better understanding of the distinction between temporal processing and cognitive control functions made in the manuscript. 

3-Third, the assessment of cognitive functions using the MMSE screening as a single measure does not seem broad enough to draw firm conclusions. The authors address this in the limitations already, but the conclusions drawn from the study do not reflect this limitation. I would strongly suggest to temper all claims regarding “cognitive functions” such that it becomes clear at all times that this refers to a rather superficial cognitive screening, rather than an in-depth or in any way specific assessment of cognitive functioning.

RESPONSE: We clarified on page 8 that “the MMSE represents one of the most routinely used screening tools in clinical practice, even if it provides only a generic assessment of cognitive decline”. Moreover, we hope that it is now clear that “cognitive control functions” refer to the (non-temporal) cognitive operations acknowledged in pacemaker-based models of time (see above). 

4-In addition to that, the wording sometimes obscures the exact comparisons that were being made in the paper. Sometimes, I had the impression that the authors aimed at comparing timing abilities between two dissimilar groups (healthy vs. “pathological” aging), but the analyses refer to the entire sample. I believe that this is preferable, given that group comparisons would not be powerful enough, but the wording should be adapted accordingly, also in the Manuscript Title. 

RESPONSE: We implemented this suggestion throughout the manuscript (e.g., the use of “vs.” was deleted and substituted with expressions such as “much older participants and more compromised participants”). The title was also amended as follows: “Explicit and implicit timing in older adults: Dissociable associations with age and cognitive decline”. 

5-In addition, the word “aging” seems inappropriate here, as there was no longitudinal or group comparison, but this was instead a cross-sectional study in a population of older adults. This should be made very clear throughout the text.

RESPONSE: Thank you for the useful clarification. The word aging has been deleted from the manuscript when used to refer to our findings. 

More detailed comments:

Abstract:

The wording in the results and discussion sections of the abstract should be adapted for clarity. For instance, “… processing of implicit, but not explicit, timing is differentially affected in healthy and pathological aging” seems to suggest that patterns of timing abilities significantly differ between healthy and non-healthy older adults.

RESPONSE: The Abstract has been almost rewritten to account for this and other comments raised by Reviewer 1. Specifically, the result and discussion sections of the Abstract have been amended as follows: “Results for the explicit timing task showed a flatter psychometric curve with increasing age or decreasing MMSE scores, pointing to a deficit at the level of cognitive control functions rather than of temporal processing. By contrast, for the implicit timing task, a decrease in the MMSE scores was associated with a reduced foreperiod effect, an index of implicit time processing. Overall, these findings extend previous studies on explicit and implicit timing in healthy aged samples by dissociating between age and cognitive decline (in the normal-to-pathological continuum) in older adults”.

Introduction:

1-The concept of “temporal processing” should be introduced in more detail. What does it entail, and from what related concepts does it differ? How is temporal processing distinguished from cognitive functioning in general, if at all (see above)?

RESPONSE: We thank the reviewer for this useful comment that allowed us to better frame the concept of temporal processing. At the very outset of the manuscript, we changed the first line of the Introduction by stating that: “Age-related changes in the performance of temporal tasks within the millisecond-to-second range intervals are commonly reported”. This way, it is clear that we are referring to the processing of temporal information in this specific temporal range. We took it for granted in the previous version of the manuscript. Moreover, as reported in our previous responses, we specified what “temporal processing” means within the framework of pacemaker-based models of time. 

2-Paragraph starting with “To our knowledge, … “: The second sentence referring to previous studies is somewhat ambiguous and should be revised.

RESPONSE: The sentence (page 7) has been amended as follows: “To our knowledge, explicit and implicit timing have been thus far compared in healthy older adults only”. Moreover, the paragraph referring to previous studies has been amended to improve clarity and readability (“As an example, Droit-Volet and colleagues (2019) devised a between-participants design, in which one group of older adults and one group of younger adults performed an explicit timing task (i.e., temporal generalization task), whereas different groups of older and younger adults were engaged in an implicit timing task (i.e., a variant of the foreperiod task; Piras & Coull, 2011). Participants’ performances on explicit and implicit timing tasks were also correlated to cognitive scores derived from neuropsychological tests)”. 

3-Same paragraph, last portion: It is not valid to conclude that lower attentional abilities are in any way mechanistically related to temporal processing, please see the paper by Borsboom that I recommended above. This must be revised.

RESPONSE: We understand the point raised by the reviewer. When we presented the study by Droit-Volet and colleagues (2019), we tried to report their conclusions as faithfully as possible. Indeed, the authors explained their results by stating that “the variability of duration judgements was greater in older than young participants, though this was directly related to older participants’ lower attentional capacity”. However, in the light of the reviewer’s comments, we deleted the paragraph stating that “performance of older participants was a consequence of their lower attentional abilities”. We now state (page 7) that “older participants were more variable than younger ones in the explicit timing task and their performance was explained by lower attentional capacity rather than age”.

4-Same paragraph, last sentence + first sentence following paragraph: Is this a statistically meaningful difference? Is it possible that cognitive scores were just not sensible enough to show an effect, e.g. due to the typically low reliability of cognitive measures?

RESPONSE: As said above, in this introductory section, we are reporting previous findings as originally presented by the authors. In particular, Droit-Volet and colleagues (2019) performed hierarchical regression analyses to determine whether age or score on the cognitive tasks was the best predictor of performance. Accordingly, although it is possible that cognitive measures had low reliability, we think it would be unfair for the authors of that previous study to question their results as they have already been peer-reviewed and published. In any case, we amended the sentence as follows: “By contrast, older adults showed a greater reliance on the hazard function than younger adults, a result that was significantly associated with age but not cognitive scores”.

5-Paragraph starting with “Collectively, …”, third sentence: The current design is implied to provide more direct evidence for the effect of interest than previous studies, but this is not the case, as it relies on the same correlational logic, rather than on a true experimental manipulation of “age” and “cognition”. This is not a problem per se (or at least not specific to this particular paper), but it should be discussed accordingly.

RESPONSE: We agree with the reviewer. Please note that a similar concern was also raised by Reviewer 1 (point 3). We now amended this part as follows (page 8): “Collectively, the correlational findings by Droit-Volet and colleagues (2019) speak to different possible influences of age and cognitive functions in explicit and implicit timing tasks. In the present study, we aimed to advance our knowledge about performance of older adults on explicit and implicit timing tasks by considering not only healthy but also pathological older participants. This allowed capturing differences in explicit and implicit timing tasks linked to age and pathological cognitive decline, two variables that, although often correlated, are not systematically associated (Jansen et al., 2018)”. 

6-Paragraph starting with “Different, …”, third sentence: Remove the comma before “which”, as it changes the meaning of the sentence.

RESPONSE: Please note that this paragraph about the predictions of the study was amended. It now reads as follows: “Concerning implicit timing, we expected to find a significant association of performance on the foreperiod task with age rather than with MMSE scores, replicating previous research (Droit-Volet et al., 2019). As concerns explicit timing, if the poor performance of older participants on the time bisection task depends on a deficit in temporal processing, this should be reflected by a (rightward) shift of the psychometric curve. According to previous literature reporting a slowing down of the pacemaker with age (see Turgeon, Lustig, & Meck, 2016), we predicted to find a significant association between the rightward shift of the curve and age. If, conversely, changes in the performance of older participants depend on their reduced cognitive control functions, this should translate into a flatter psychometric curve. Therefore, we predicted a significant association between MMSE scores and the flattening of the curve”. 

7-Same paragraph, last sentence: It seems very difficult to show that two variables are “unrelated”, as this implies acceptance of the null hypothesis. Please revise the wording.

RESPONSE: Revised accordingly. Specifically, the word “unrelated” has been deleted and the predictions have been framed taking into account the comments of both reviewers (see our response above). 

Methods:

As a general suggestion, I would recommend to walk the reader more gently through the analyses and explain which particular questions are answered by which tests and what is implied by which outcome.

RESPONSE: We implemented this useful suggestion and added the following explanations on pages 11 and 12: “For the interpretation of the model terms, a significant main effect of MMSE score would indicate a change in the intercept value (since all variables were centered, the intercept is the expected value of the logistic curve at the middle interval duration, i.e., 1200 ms, when MMSE and AGE variables are at their mean value) for a 1 unit change in the MMSE score. As can be appreciated from Figure 1B, the higher the value of the psychometric curve at the middle interval duration, the higher the shift of the curve towards the left (i.e., over-estimation), and vice versa. In sum, a significant main effect of MMSE with an odds ratio greater than 1 would indicate a progressive shift of the curve towards the left with increasing MMSE score (if lower than 1, this would indicate a progressive shift of the curve towards the right with increasing MMSE score). The same logic applies to the main effect of AGE. The flattening of the curve represented in Figure 1B is captured by the interaction of MMSE score (or Age) and Interval duration. A significant odds ratio greater than 1 would indicate a significant steeping of the curve with the increase of the MMSE variable (or Age), whereas a significant odds ratio lower than 1 would indicate a significant flattening of the curve with the increase of the MMSE variable (or Age)”. Moreover, for the implicit timing task, we added that: “As explained above, the foreperiod effect is the well-observed lowering of RT with increasing interval duration. This effect is captured by the negative slope of the regression line (see Figure 2B). A significant negative interaction effect would indicate, hence, a stronger foreperiod effect with increasing MMSE score (or Age). On the contrary, a significant positive interaction would indicate a progressive reduction of the foreperiod effect with increasing in the variable (MMSE or Age)”.

General comment: MMSE-scores are not independent of participant age. Does this represent a problem for the chosen analysis approach? Consider adapting the analysis strategy, but at least, this issue should be discussed.

RESPONSE: As already reported in the previous version of the manuscript (page 9), MMSE scores were corrected for age and level of education as suggested by Magni et al. 1996 for the Italian population. Moreover, although the analysis considers the unique (i.e., not shared) variance explained by each predictor, the correlation between MMSE and Age variables was negligible (r = .0676, R2 = .0046). This information is now reported in the revised version (page 11): “The correlation between Age and MMSE score variables was very low (r = -.068)”.

Regarding the strategy to deal with no-response trials: Did I understand correctly that no-response trials were included in the original analyses? Why were they not discarded on a trial-by-trial basis in that analysis, rather than discarding an entire case in a supplementary analysis? Would the results change when excluding no-response trials?

RESPONSE: We apologize for having forgotten to report that for the explicit timing task “data from trials with missing responses were discarded from the analysis” (now page 11). Moreover, for the implicit timing task, “data from error trials, i.e., anticipated (< 100 ms) or missing responses to the target, were not included” (page 12)”. To check for the robustness of our results, in a second step, we repeated the analyses by excluding participants according to the proportion of trials in which they did not provide a response. 

Results:

Paragraph “Implicit timing task”, second sentence: Could you please explain in more detail what it means that “the model was a bit stressed”?

RESPONSE: We apologise if the sentence was unclear. In the revised version, the sentence was rephrased as follows (page 15): “Visual inspection of the residuals showed that they were skewed”.

Same paragraph, last sentence: This is very difficult to see from the Figure. Please provide some more guidance or adapt accordingly.

RESPONSE: A similar concern was also raised by Reviewer 1. As replied above, we understand that the figure doesn’t “clearly” show the strength of the FP effect in a very intuitive way. Unfortunately, however, there are no other illustrative ways to present the data as have been analyzed here. We hope that the explanation added on page 12 (“As explained above, the foreperiod effect is the well-observed lowering of RT with increasing interval duration. This effect is captured by the negative slope of the regression line (see Figure 2B). A significant negative interaction effect would indicate, hence, a stronger foreperiod effect with increasing MMSE score (or Age). On the contrary, a significant positive interaction would indicate a progressive reduction of the foreperiod effect with increasing in the variable (MMSE or Age))” is sufficient to better understand the Figure. 

Discussion:

First paragraph: The statement “in both healthy and pathological aged populations” is misleading; cf. my earlier comment. Please revise.

RESPONSE: This statement now reads as follows: “In this correlational study, we compared explicit and implicit timing in both healthy and pathological older participants, thus, extending recent research on explicit and implicit timing in healthy older adults (Droit-Volet et al., 2019).

Second paragraph, last sentence: The term “pathological” cognitive decline is not appropriate here, given that (1) the classification as “pathology” was merely based on a screening tool rather than on a solid clinical diagnosis, and (2) it was not part of an actual comparison. Please rephrase.

RESPONSE: As now reported in the manuscript, unhealthy participants received a diagnosis of either MCI or dementia. Accordingly, we hope that the term pathological is now justified in the manuscript. However, we understand the reviewer’s suggestion, and the expression “pathological cognitive decline” was deleted from the sentence, which now reads as follows: “Therefore, much older and more compromised participants made less precise temporal judgments in the time bisection task”.

Third paragraph, sentence starting with “Another possible, …”: It is suggested here that attentional deficits might explain worse performance in relatively older adults and in relative low performers on the MMSE on the explicit timing task. Such a deficit is not definitely shown, as there was no specific test to measure attention. Even if an attention deficit had been shown to coincide with worse explicit timing performance, it should be made explicit that this does in no way allow for a mechanistic or causal interpretation of the results. Finally, if attention were indeed causally related to timing abilities, why would this effect be limited to the explicit timing task and not also the implicit timing task?

RESPONSE: We agree with the reviewer and removed the speculation about the involvement of attention from the manuscript. 

Same paragraph, sentence starting with “In any case, …”: This wording suggests that age and cognitive performance independently modulate task performance, but they are confounded. Please rephrase, if applicable (cf. my earlier comment regarding the Methods).

RESPONSE: This paragraph was deleted (see also below). 

Same paragraph, sentence starting with “Remarkably, …”: This sentence should be rephrased. First, it is difficult to conclude that “neither age nor cognitive impairment altered timekeeping mechanisms”, as this implies the acceptance of the null hypothesis. Second, it should not be stated that either of those variables “influenced” cognitive processes to avoid suggesting a mechanistic relationship.

RESPONSE: This paragraph now reads as follows (see page 18): “Overall, findings from the explicit time bisection task point to a deficit for much older or more compromised participants at the level of cognitive control functions rather than of temporal processing, as evinced by a flatter psychometric curve and the lack of a systematic under- or over-estimation of interval durations”.

Fourth paragraph, sentence starting with “Considering that implicit timing poses…”: Can it actually be stated that implicit timing poses less demands on cognitive processing as compared to explicit timing? What is the basis for this claim?

RESPONSE: We specified (page 19) that: “For the type of instructions given to participants and the task goal itself, implicit timing is believed to pose fewer demands on cognitive processes as compared to explicit timing”. This claim is based on the literature cited in the manuscript (e.g., Coull & Nobre, 2008) that divides timing tasks, and underlying temporal processes, into explicit and implicit as a function of the type of instructions given to participants. In explicit timing tasks, such as the time bisection task also employed in our study, the task goal itself is temporal, as participants are explicitly informed about the need to pay attention to elapsing time and to memorize some specific duration. It follows then that under this condition, performance on the explicit timing task is also related to the participants’ ability to correctly process interval durations and to maintain them in memory for making a subsequent decision (e.g., shorter or longer than the standard durations?).

By contrast, in implicit timing tasks, like the foreperiod task, the task goal itself is not temporal and no explicit requirements to process time are provided. Accordingly, it is assumed that implicit timing tasks pose fewer demands on cognitive processes as compared to explicit timing tasks. 

Paragraph starting with “The presence of a larger…”, last sentence: Revise the wording in this sentence. It seems to suggest a comparison between “older”/ “healthy” and “impaired” / “pathological” adults (cf. my previous comments).

RESPONSE: The sentence (see page 19) now reads as follows: “The presence of a larger foreperiod effect in much older participants is explained by the fact that they were slower than less older participants at shorter interval durations having, in turn, more room for improvement at longer durations”.

Paragraph starting with “Taken together, …”, third sentence: Please remove the statement regarding early markers. This cross-sectional design does not allow for the conclusion of any longitudinal effects. In addition, I understood that the authors conceptualized temporal processing to be distinct from cognitive functioning. How can this be reconciled with the viewpoint that temporal processing can serve as a marker for cognitive changes?

RESPONSE: Removed as suggested. 

Same paragraph, fourth sentence: If this paradigm poses “very little demands” on cognitive resources, then why is it worth to assess a relationship with MMSE scores at all?

RESPONSE: Although implicit timing tasks are believed to pose fewer demands on performance than explicit timing tasks, they still require an elaboration of temporal information. Accordingly, this temporal part deserves to be investigated in relation to cognitive decline. 

Same paragraph, fifth sentence: This wording implies a direct statistical comparison between explicit and implicit timing tasks, please revise. Similarly, a few sentences further, it is stated that the MMSE was “enough sensitive to differentiate performance on explicit and implicit tasks”, but this, again, would need to be shown statistically.

RESPONSE: The statement “enough sensitive to differentiate performance on explicit and implicit tasks” was removed from the manuscript. 

Last paragraph, second sentence: It seems difficult to make conclusions regarding “age-related” changes based on the present data, as the study population were older adults exclusively.

RESPONSE: We avoided using the wording “age-related” changes when referring to our findings throughout the manuscript. 

6. PLOS authors have the option to publish the peer review history of their article (what does this mean?). If published, this will include your full peer review and any attached files.

Do you want your identity to be public for this peer review? For information about this choice, including consent withdrawal, please see our Privacy Policy.

Reviewer #1: No

Reviewer #2: No

---

## [Decision Letter · Decision Letter 1]

21 Feb 2022

Explicit and implicit timing in older adults: Dissociable associations with age and cognitive decline

PONE-D-21-29634R1

Dear Dr. Mioni,

We’re pleased to inform you that your manuscript has been judged scientifically suitable for publication and will be formally accepted for publication once it meets all outstanding technical requirements.

Kind regards,

Bradley R. King

Academic Editor

PLOS ONE

Additional Editor Comments (optional):

Reviewers' comments:

Reviewer's Responses to Questions

**Comments to the Author**

1. If the authors have adequately addressed your comments raised in a previous round of review and you feel that this manuscript is now acceptable for publication, you may indicate that here to bypass the “Comments to the Author” section, enter your conflict of interest statement in the “Confidential to Editor” section, and submit your "Accept" recommendation.

Reviewer #1: All comments have been addressed

Reviewer #2: All comments have been addressed

2. Is the manuscript technically sound, and do the data support the conclusions?

Reviewer #1: (No Response)

Reviewer #2: Yes

3. Has the statistical analysis been performed appropriately and rigorously? 

Reviewer #1: (No Response)

Reviewer #2: Yes

4. Have the authors made all data underlying the findings in their manuscript fully available?

Reviewer #1: (No Response)

Reviewer #2: Yes

5. Is the manuscript presented in an intelligible fashion and written in standard English?

Reviewer #1: (No Response)

Reviewer #2: Yes

6. Review Comments to the Author

Reviewer #1: (No Response)

Reviewer #2: I would like to thank the authors for their thorough revision of the manuscript. I have no further comments.

7. PLOS authors have the option to publish the peer review history of their article (what does this mean?). If published, this will include your full peer review and any attached files.

Reviewer #1: No

Reviewer #2: No

---

## [Editor Report · Acceptance letter]

1 Mar 2022

PONE-D-21-29634R1 

Explicit and implicit timing in older adults: Dissociable associations with age and cognitive decline 

Dear Dr. Mioni:

I'm pleased to inform you that your manuscript has been deemed suitable for publication in PLOS ONE. Congratulations! Your manuscript is now with our production department. 

Kind regards, 

on behalf of

Dr. Bradley R. King 

Academic Editor

PLOS ONE